

# Effects of media multitasking frequency on a novel volitional multitasking paradigm

Jesus J. Lopez[1] and Joseph M. Orr[1,2]

[1] Department of Psychological and Brain Sciences, Texas A&M University, College Station, TX, United States of America
[2] Institute for Neuroscience, Texas A&M University, College Station, TX, United States of America

## ABSTRACT

The effect of media multitasking (*e.g.,* listening to podcasts while studying) on cognitive processes has seen mixed results thus far. To date, the tasks used in the literature to study this phenomenon have been classical paradigms primarily used to examine processes such as working memory. While perfectly valid on their own, these paradigms do not approximate a real-world volitional multitasking environment. To remedy this, as well as attempt to further validate previously found effects in the literature, we designed a novel experimental framework that mimics a desktop computer environment where a "popup" associated with a secondary task would occasionally appear. Participants could choose to attend to the popup, or to ignore it. Attending to the popup would prompt a word stem completion task, while ignoring it would continue the primary math problem verification task. We predicted that individuals who are more impulsive, more frequent media multitaskers, and individuals who prefer to multitask (quantified with the Barratt Impulsiveness Scale, a modified version of the Media Use Questionnaire, and the Multitasking Preference Inventory) would be more distracted by popups, choose to switch tasks more often and more quickly, and be slower to return to the primary task compared to those who media multitask to a lesser degree. We found that as individuals media multitask to a greater extent, they are slower to return to the previous (primary) task set and are slower to complete the primary task overall whether a popup was present or not, among other task performance measures. We found a similar pattern of effects within individuals who prefer to multitask. Our findings suggest that overall, more frequent media multitaskers show a marginal decrease in task performance, as do preferential multitaskers. Attentional impulsivity was not found to influence any task performance measures, but was positively related to a preference for multitasking. While our findings may lack generalizability due to the modifications to the Media Use Questionnaire, and this initial study is statically underpowered, this paradigm is a crucial first step in establishing a more ecologically valid method to study real-world multitasking.

Corresponding author
Joseph M. Orr, joseph.orr@tamu.edu

## INTRODUCTION

The preponderance of information available at our fingertips makes multitasking seem like the norm. Unsurprisingly, the proportion of time an individual multitasks with multiple information sources increased 10% from 6 h and 20 min a day, to 7 h and 38 min a day between 1999 and 2009 (*Rideout et al., 2010*). Furthermore, research suggests some negative impacts of screen time (*i.e.,* time spent viewing television, phone/tablet, or laptop), on cognitive abilities and other psychosocial factors, and particularly on the development of these functions (*Domingues-Montanari, 2017*; *Hooghe & Oser, 2015*; *Sigman, 2012*). As such, it is critical to understand the costs and potential benefits of frequent media multitasking, often defined as the simultaneous use of two or more media types or the act of quickly switching between different media types (*Minear et al., 2013*).

### Previously found effects of media multitasking

To that end, research has aimed to establish differences in information processing as a function of time spent media multitasking, with a typical focus on extreme groups comparisons. A number of studies have now identified a negative association between media multitasking frequency and performance on cognitive tasks that require focus and cognitive stability such as distractor filtering (*Lottridge et al., 2015*; *Moisala et al., 2016*; *Murphy & Creux, 2021*; *Wiradhany & Nieuwenstein, 2017*), inhibitory control (*Baumgartner et al., 2014*; *Schutten, Stokes & Arnell, 2017*), and sustained attention (*Ralph & Smilek, 2017*; *Ralph et al., 2014*). Thus far, heavy media multitasking frequency has been linked to deficits in single task settings, but research into domains where one might expect multitaskers to excel, such as task switching, has produced more mixed results. For example, Ophir and colleagues (*2009*) found a negative association between heavy media multitaskers and task switching, while *Alzahabi & Becker (2013)* found the opposite relationship. Indeed, a growing body of work suggests no relationship between media multitasking and task switching performance (*Baumgartner et al., 2014*; *Minear et al., 2013*). More recently, *Rogobete, Ionescu & Miclea (2021)* found that no linear relationship of media multitasking on task switching, but, when comparing extreme groups, the heavier media multitaskers counterintuitively performed better than low media multitaskers. Given these mixed results, more insight is necessary to describe the effect media multitasking has on this aspect of executive function.

The Media Use Questionnaire (MUQ) was developed by Ophir and colleagues (*2009*) to quantify the amount of time an individual media multitasks during a typical media-consumption hour. Participants are first asked how many hours a week they use different media sources, followed by how often they concurrently use each other media type. From this information it is possible to quantify an individual's Media Multitasking Index (MMI). Across a variety of executive function tasks, Ophir and colleagues found that heavy media multitaskers performed worse compared to those who multitask less. With this in mind, Ophir and colleagues suggested that heavy media multitaskers are less able to filter out irrelevant information when compared to their lighter media multitasking counterparts. In line with Ophir and colleagues, a number of studies have now shown a similar pattern

of results (*Cain & Mitroff, 2011*; *Cardoso-Leite et al., 2016*; *Heathcote et al., 2014*; *Lottridge et al., 2015*; *Wiradhany & Nieuwenstein, 2017*).

Can propensity for media multitasking be predicted by individual emotional or attitudinal differences? Previous work indicates that sensation-seeking and impulsivity might influence the frequency of media multitasking. For example, research suggests that a weak positive association between total multitasking use and sensation seeking ratings on the Brief Sensation Seeking Scale exists (*Jeong & Fishbein, 2007*). Similarly, sensation seeking has also been found to predict media multitasking frequency as measured by the MMI (*Kononova, 2013*). Sanbonmatsu and colleagues (*2013*) found that individuals with higher MMI scores also scored high on impulsivity, and moreover, performed worse on the Operation Span Task, a complex span task that involves rapid task switching, or multitasking, as defined by *Madore & Wagner (2019)*. Furthermore, media multitasking has been found to be associated with attentional impulsivity, as measured by both performance on a Go/No-Go task and a subscale of the Barratt Impulsiveness Scale (BIS), as well as lower self-reported initiatory self-control (*Shin, Webb & Kemps, 2019*). Finally, Minear and colleagues (*2013*) found that heavy media multitaskers reported being more impulsive while also showing worse performance on measures of fluid intelligence. Taken together, these findings point towards the possibility of an emotional and cognitive basis behind this phenomenon. However, some research does suggest that the effects of screen time, at least in regard to adolescent well-being, have thus far been overstated and are in fact, much smaller than has been purported (*Orben & Przybylski, 2019*).

Although most studies suggest a negative relationship between media multitasking frequency and cognitive performance, a number have found no difference associated with media multitasking use, or even findings in the opposite direction. Indeed, a meta-analysis by *Wiradhany & Nieuwenstein (2017)* found a weak association between media multitasking and distractibility and a more recent meta-analysis by *Parry & Le Roux (2021)* found a weak association between media multitasking and general cognitive function. These weak patterns of effects are prevalent in the task switching (*Alzahabi & Becker, 2013*; *Alzahabi, Becker & Hambrick, 2017*; *Schneider & Chun, 2021*), dual tasking, (*Ie et al., 2012*), and inhibition literature (*Rogobete, Ionescu & Miclea, 2021*) . Interestingly, two studies have found that intermediate or moderate multitaskers show better N-back performance compared to heavy and light media multitaskers (*Cardoso-Leite et al., 2016*; *Shin, Linke & Kemps, 2020*). Nevertheless, research examining intermediate or average media multitaskers is much less common than the extreme groups comparisons that the literature has to date focused on.

## Purpose

In summary, there are still many outstanding questions regarding media multitasking's effect on task performance. Though the literature has found some effects, these have been derived from already established paradigms that have been historically used to study other cognitive processes that are not always immediately reminiscent of multitasking. In day-to-day life, multitasking is usually done at the leisure of the individual, with task switches occurring randomly and sporadically; this is counter to most lab-based studies of

multitasking, in which the experimenter dictates when and how an individual multitasks. By giving participants the choice of when to switch to a secondary task, as well as modeling the task to be more similar to a multitasking environment, we can examine whether media multitasking frequency relates to one's tendency to switch tasks often as well as overall task performance. Thus, in the current study, we developed a novel experimental framework more analogous to multitasking in day-to-day life by having participants complete a primary, monotonous task with sporadic "interruptions" presented in the form of an opportunity to switch to a different, secondary task. We hope to use this paradigm to dispel the ambiguity in the current literature in the field by allowing us to more closely examine the differences between individuals' task performance and the effect extensive daily media multitasking may have on it by using a task specifically designed to emulate real-world multitasking. The ability to replicate previously established effects with this more ecologically valid paradigm would provide further support for those effects as well. Further, a majority of the literature has focused on an extreme groups approach. While this is obviously very valuable information to have, the question still remains as to whether any degree of media multitasking can affect task performance and not only in extreme "high" or "low" cases. The current work seeks to reconcile the limitations of the extreme groups approach, as well as establish a more ecologically valid task paradigm that can then be used to further examine cognitive differences and how they are affected by media multitasking.

In the current study, we operationalized the act of multitasking as the attempt to perform more than one task concurrently, resulting in the act of switching back and forth between tasks (*Madore & Wagner, 2019*). To that end, we designed the framework of our novel paradigm around the Operation Span Task (OSPAN) devised by *Turner & Engle (1989)* as it requires participants to complete two tasks concurrently. In fact, Sanbonmatsu and colleagues previously used the OSPAN to examine multitasking ability (*Sanbonmatsu et al., 2013*). In our paradigm, a participants' primary task was a math problem verification task, similar to the OSPAN. However, in some trials, a pop-up message would appear, which asked if the participant wanted to switch tasks. The pop-up prompts were implemented to be reminiscent of the notifications that appear on our phones and computers and appeared randomly throughout a block of trials. If the participant indicated that they wanted to switch, they were then given a word stem completion to solve, after which they returned to the primary task. This is another differentiation from the OSPAN, as the secondary task in that paradigm is not optional and indexes an individual's working memory by asking participants to recall a series of letters that are presented after each primary task trial at the end of a block. In our current task, the participant does not have to hold any objects in their working memory as they work through the task, instead indexing their ability to task-switch.

Our paradigm also draws from the voluntary task switching (VTS) literature. We chose to model our paradigm from this literature because of the similar scenarios that are presented to participants in those paradigms. Here, the volitional aspect that is common in VTS tasks paradigms is present, albeit with some fundamental changes. The participant is only prompted to respond on a random subset of trials as opposed to having the option during each trial, a deviation from most VTS paradigms (*Arrington & Logan, 2004*; *Mayr &*

*Bell, 2006*; *Orr & Weissman, 2011*). For every trial in which the option to switch tasks is not presented, the participant is only able to complete the primary task. Again, this was done in an attempt to further emulate a scenario in which real-world multitasking might occur. For example, an individual may be focused on a task on their computer, when a random popup in the corner of the screen may catch their eye. The individual then has the option to switch tasks away from their main focus to attend to this popup, a crucial element that is not present in VTS paradigms.

## Hypotheses

We predicted a positive relationship between MMI score and the rate at which participants would elect to switch to the secondary task (*Switch Rate*). We also expected that participants would show a "*Return Cost*", *i.e.,* respond slower to return to the primary task following a switch to the secondary task that would be positively predicted by media multitasking in line with the suggestion that media multitasking frequency is associated with decreased executive function (*Baumgartner et al., 2014*; *Cain et al., 2016*). Additionally, we predicted that individuals who media multitask more often would choose to switch to the secondary task more quickly (in the form of a faster time to elect to switch tasks on relevant trials, which we refer to as *Popup$_{select}$*). In line with the suggestion that frequent multitaskers show increased difficulties with distractor filtering, we predicted that MMI score would also show a positive relationship with the amount of interference exhibited on trials where a pop-up was presented, but the secondary task wasn't chosen (*Interference Cost*).

## MATERIALS & METHODS

### Participants

A total of 90 participants (62 female, 28 male) with ages ranging from 18–23 years ($M = 19.15$, $SD = 0.9$) fully completed the procedure. Two participants were dropped due to non-completion of the study. Participants were recruited from the Texas A&M Psychology Subject Pool and received course credit for participating. No target sample size was determined, with the intent to collect as much data as possible through the course of a full semester. A post-hoc power analysis was performed, described below. Demographic information is reported in Table 1. Participants were not prescreened for media multitasking frequency and only had to be English-speakers who were right-handed, neurotypical, had full color vision, and were between the ages of 18-30 years old; in addition, participants were not told this was a study on multitasking until they were consented to participate in the study. Study procedures were deemed exempt from the requirements of the Common Rule (45 CFR 46.101[b]) by the Texas A&M Institutional Review Board, approval reference number IRB2018-1456M. The authors confirm that we have reported all measures, conditions, data exclusions, and the method of sample size determination.

### Media multitasking index

A Media Multitasking Index (MMI) was calculated in order to assess the degree to which participants multitask with different forms of media (*Ophir, Nass & Wagner, 2009*). Participants first completed the Media Use Questionnaire, which asked participants to

**Table 1  Main survey descriptive statistics.** Descriptive statistics for the MUQ (Media Use Questionnaire), MPI (Multitasking Preference Inventory), BIS-11 (Barratt's Impulsiveness Scale), and the three second order factors within the BIS. Note that the MUQ scores are operationalized as a Media Multitasking Index (MMI) and are presented as such in this table.

|  | MMI score | MPI score | Total BIS | Attentional | Motor | Nonplanning |
|---|---|---|---|---|---|---|
| Mean | 2.95 | 38.54 | 62.59 | 17.76 | 21.1 | 23.73 |
| Median | 2.82 | 37 | 61 | 17 | 20 | 23.5 |
| SD | 1.28 | 10.93 | 9.72 | 3.86 | 4.29 | 4.17 |

estimate how many hours per week they use each individual form of media (using a sliding scale ranging from 0–80 (in hours). They were then given a matrix asking, for each media type they use, how often they concurrently used each of the other mediums using a 5-point Likert scale ("Always," "Most of the time," "Some of the time," "A little of the time," or "Never"). Although these values are not disclosed to the participants, numeric values were assigned to each of the matrix answers, such that "1.0" represented "Always", "0.75" corresponded to "Most of the time," "0.5" to "About half the time", "0.25" to "Sometimes," and "0" to "Never." The sum of these values across each primary medium use weighted by the percentage of time spent with the corresponding primary medium was then computed to yield a participant's Media Multitasking Index (MMI) score. This final MMI score can be interpreted as the level of media multitasking the participant is engaged in during a typical media-consumption hour so that the higher the MMI, the greater the amount of time that participant spends media multitasking in an hour. Figure S1 shows the equation Ophir and colleagues (*2009*) used for calculating MMI scores, again used in the current study.

The original version of the Media Use Questionnaire (*Ophir, Nass & Wagner, 2009*) was modified for the current study to reflect current trends in media usage. This modified version assessed 12 media types; computer-based applications (*e.g.*, word processing, excel), web surfing (not including social media sites), text-based media such as print books, ebooks, magazines, newspapers (for school/work/pleasure), television programs (TV based or online streaming), streaming videos (*e.g.*, YouTube, BuzzFeed, other short clips), listening to music, listening to nonmusic audio (*e.g.*, audio books, podcasts, talk radio, *etc.*), video based games (console, computer, phone/tablet based), voice calls (landline, cellphone, skype), reading/writing emails, viewing social media (facebook, instagram, snapchat, twitter, *etc.*), and "other" media types. The original version of the questionnaire's "instant messaging" media type was replaced with "social media" to reflect the rise of social media and the decline of instant messaging since the creation of the questionnaire. We also changed the wording for several media types. "Print media" was changed to "text media" to reflect the popularity of e-readers, "telephone and mobile phone voice calls" was changed to "voice calls," "computer-based video" was renamed to "streaming video" (to reflect current trends towards services such as YouTube, Netflix, and Hulu), and "video or computer games" was renamed to "video games." Ophir et al.'s version of the index used only a 4-point Likert scale ranging from "Most of the time" to "Never". We added the additional answer choice of "Always" in an attempt to get a more precise measure of

media multitasking occurrence. The addition of the extra choice of "Always" was done to counterbalance the already existing "Never" answer choice.

## Multitasking Preference Inventory (MPI)

Participants also completed the Multitasking Preference Inventory, a 14-item questionnaire devised by *Poposki & Oswald (2010)* to index an individual's general "preference towards multitasking." It consists of fourteen statements relating to their opinions on performing tasks (ex: "I prefer to work on several projects in a day, rather than completing one project and then switching to another.") that they then indicate on a 1 (*Strongly Disagree*) to 5 (*Strongly Agree)* Likert scale as to how well each describes them. Scoring was done in accordance with *Poposki & Oswald (2010)* and includes the summation of all items once the appropriate questions have been reverse scored. Higher scores on this measure suggest a higher inclination to want to multitask.

## Barratt Impulsiveness Scale-11 (BIS-11)

Too assess impulsivity, the BIS-11 was administered and scored according to previous works, consisting of the sum of all items following the reverse scoring of the appropriate questions (*Patton, Stanford & Barratt, 1995*). The questionnaire consists of 30 items on a 1 (*Rarely/Never*) to 4 (*Almost Always/Always*) Likert scale related to impulsive behaviors and attitudes. The scale can be further broken down into 6 first order factors (Attention, Cognitive Instability, Motor, Perseverance, Self-Control, and Cognitive Complexity) and 3 s order factors (Attentional, Motor, and Nonplanning). Following *Sanbonmatsu et al. (2013)*, all 30 questions of the BIS-11 were used, with the Attentional impulsivity sub-scale being especially of interest for the current study.

## Multitasking paradigm

Figure 1 is a representation of the multitasking paradigm. Participants completed the multitasking task, created using PsychoPy version 3.0.6 (*Peirce et al., 2019*). All participants completed the task on a 21-inch, with default monitor settings as defined by PsychoPy. All monitor settings were determined by the default test monitor settings within PsychoPy. For the primary task, participants checked the validity of math operations (*e.g.*, '$(3 - 2) \times 1 = 4$') *via* key press, with "C" indicating the math problem was correct, and "I" to indicate an incorrect problem. The math operations were on the screen for 5 s. Participants were informed that a correct response to the primary task was worth 3 points. Incorrect or responses not made in time would deduct this same amount from the total. Participants were shown their running total after every trial. The points did not have a monetary value, but to incentivize participants to achieve as high a score as possible, they were shown a "high score" at the end of each block. This high score was the same for each participant.

On one out of every six primary task trials, a text "popup" would appear on the screen 500 ms after the primary task appeared, reading "A New Task is Available! Press 'Y' to switch tasks". This popup would appear on the top right corner of the screen. The position of the popups was chosen so that they would be reminiscent of the notifications seen on computers and cell phones. The popup would appear on screen for 2 s, after which the text would disappear. Participants could choose to continue attending to the primary task

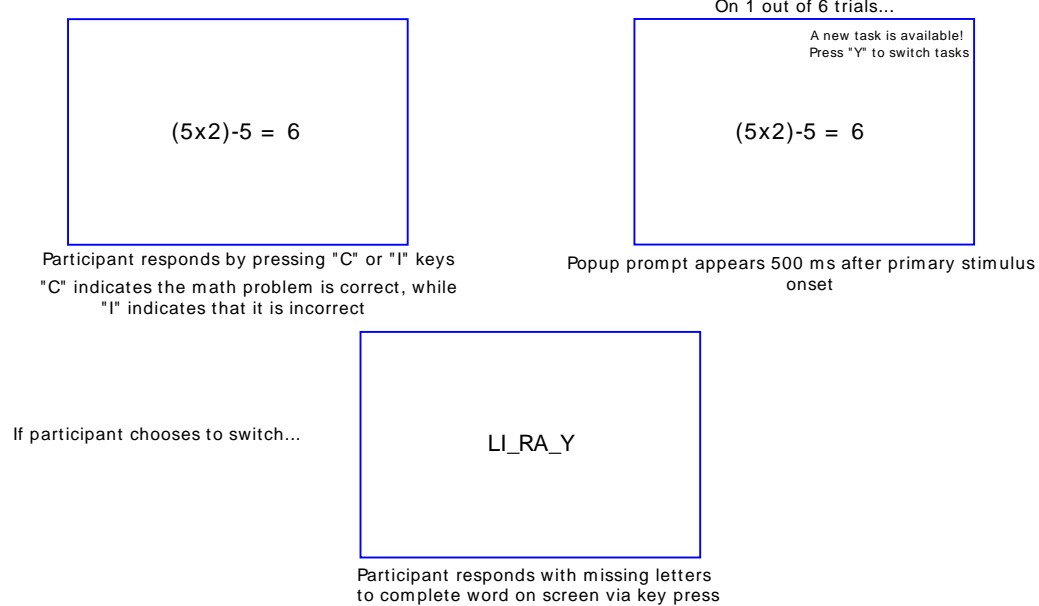

**Figure 1 Novel multitasking paradigm used.**

instead of the popup. If participants chose to switch tasks, they would then be shown a word fragment with two letters missing (*e.g.*, "HI_TO_Y"). Participants would then indicate which letters were missing *via* key press. A correct response to the secondary task was worth 10 points, with an incorrect or response not made in time would deduct this same amount from the total. The discrepancy of possible points between the primary and secondary tasks was implemented to make the secondary task more enticing and encourage multitasking, due to the greater amount of points possible for successfully completing it. Participants again were shown their running total at the end of each trial.

The task consisted of eight blocks of 20 trials each for a total of 160 trials. The number of blocks was chosen so that the task would be broken up into intervals allowing the participant to take breaks regularly while still being able to complete the experiment in under an hour. The number of pop-ups was not consistent across participants due to the randomization procedure, with an average of 27.8 ($SD = 4.4$) pop-ups per participant.

## Procedure

After providing consent *via* a written consent form, participants completed an online version of the Media Use Questionnaire, the MPI, the BIS-11, and a demographics questionnaire. After completion of the surveys, participants then completed a short practice version of the novel multitasking paradigm, followed by the full version of the task. The practice version of the task consisted of three distinct blocks. In the first block, participants completed five trials of only the primary task. Similarly, they completed five trials of only the secondary task in the second block. In the final practice block, participants completed six trials of the full task. Point values were identical to the full task. The total experiment duration was about 1 h.

## Data analysis

Analyses and plots were created using RStudio Version 1.2.5033 (*R Core Team, 2019*). Post hoc power analysis was conducted using G*Power (*Faul et al., 2009*). Survey scores were compared using Pearson's 2-tailed correlations. The main dependent measures for task performance were switch rate (the percentage of trials in which a participant switched tasks across all trials in which switches were possible), $Popup_{select}$, or the reaction time for individuals to elect to switch tasks on relevant trials, *Return Cost, i.e.,* the difference in average reaction time for primary tasks following a switch to the secondary task minus the average reaction time for all other primary task trials without a popup, and *Interference Cost, i.e.,* the difference in reaction time for primary task trials with a non-selected pop-up and primary task trials without a pop-up.

Because some participants did not switch at all throughout the task ($n = 23$), it was not possible to calculate some measures for the entire sample. The effects of media multitasking on task performance were analyzed using a hierarchical regression model consisting of the three main surveys (Attentional BIS, MMI, and MPI) to predict each measure of task performance. In the first step of the regression, we included only MMI Score, as that was the main construct of interest. In step 2, we then included the attentional sub-scale of the BIS score, with MPI score being added in step 3.

To maximize the amount of useable data, all trials in which a participant responded to either task were included in our analyses, unless otherwise noted. The data and materials for this experiment are available at (https://osf.io/nju8a/?view_only=27e3adfafbba48488a1bf0f7c20e1f4a). This experiment was not preregistered.

# RESULTS

## Survey results

Table 1 shows a breakdown of survey scores. Mean MMI was 2.95 (SD = 1.3), with significant deviation from normality ($W = 0.96$, $p = .01$). There was no difference in MMI between males and females, $F(1,88) = 1.77$, $p = .19$, $\eta_p^2 = 0.02$. Our mean MMI is relatively in line with that of other studies using the original MUQ questionnaire and its method of calculation devised by Ophir and colleagues (*2009*) (*Moisala et al., 2016*; *Ralph et al., 2014*; *Schneider & Chun, 2021*).

The mean MPI score was 38.5 (SD = 10.9), indicating an overall neutral preference for multitasking. This is slightly higher and less variable than previous studies that have also used this measure, suggesting that our sample had a slightly greater preference for multitasking. For example, a random sample of experienced Amazon MTurk workers resulted in an average of 38.01 (SD = 12.54) (*Lascau et al., 2019*). Relatedly, an in-person sample of university students found an average MPI score of 29.95 (SD = 8.72) (*Magen, 2017*).

Median Total BIS was 61.0 (SD = 9.7), with a median Attentional score of 17.0 (SD = 3.9), a median Motor score of 20.0 (SD = 4.3), and a median Nonplanning score of 23.5 (SD = 4.2). MMI was significantly correlated with BIS-Motor ($r = 0.27$), and MPI scores were correlated with Total BIS ($r = 0.26$) as well as BIS-Attentional ($r = 0.29$),

**Table 2** **Descriptive statistics for the main behavioral measures analyzed.** Switch rate refers to the percentage of trials in which a participant switched tasks across all relevant trials. Return cost refers to the difference in reaction time (in seconds) for primary tasks following a switch to the secondary task minus the reaction time for all other primary task trials without a popup. Interference Cost refers to the difference in reaction time (in seconds) for primary task trials with a non-selected pop-up and primary task trials without a pop-up. Finally, Popup$_{select}$ refers to the reaction time in seconds for individuals to elect to switch tasks on relevant trials.

| | Switch rate (%) | Return cost | Interference cost | Popup$_{select}$ |
|---|---|---|---|---|
| Mean | 0.31 | 0.34 s | 0.1s | 1.31 s |
| SD | 0.30 | 0.41 s | 0.22 s | 0.35 s |

BIS-Cognitive Instability ($r = .23$), BIS-Self Control ($r = .23$) and BIS-Motor ($r = 0.22$) sub-scale scores. However, MMI was not correlated with MPI ($r = -0.1$). Expectedly, all of the BIS sub-scales were correlated with Total BIS (all $r > 0.74$). Table S1 shows a correlation matrix of all surveys and behavioral measures.

## Multitasking performance

Participants performed the primary task (math problem verification) with high accuracy ($M = 94.3\%$, $CI = 91.8$–$97.1\%$) and the secondary task (word stem completion) with moderate accuracy ($M = 69.7\%$, $CI = 59.1$–$89.2\%$). Popups appeared on a median of 28 trials ($CI = 24.25$–$30.75$), and participants chose to switch to the secondary task on an average of 30.8% of pop-up trials ($CI = 22.6$–$54.7\%$). According to a Wilcoxon one sample $t$-test, this value significantly differed from 0.0 ($V = 2278$, $p < .001$, $r = 1.0$), however, 23 participants did not respond to any of the popups, with an additional 8 only responding to 1. Primary task reaction time was then analyzed in a repeated measures ANOVA with one factor with the following levels: Ignore (*i.e.,* popup was present but not responded to), Return (*i.e.,* previous trial on which the secondary task was performed), and No Popup (*i.e.,* no popup on current or previous trial). Only correct trials were included in this analysis and participants with less than 3 values in any cell were excluded, resulting in a final sample of 50 participants. There was a main effect of condition ($F(1.32, 64.7) = 24.9$, $p < 0.01$, $\eta_g^2 = 0.08$), and pairwise tests showed significant differences between No Popup and Return (*i.e.,* Return Cost), Return and Ignore, but not No Popup and Ignore (*i.e.,* Interference Cost), as shown in Fig. 2. A similar analysis was run for accuracy data (as in transformed), and no effect of condition was observed ($F(1.71, 85.72) = 1.4$, $p = 0.25$, $\eta_g^2 = 0.017$).

Next, we examined whether multitasking behavior was predicted by the multitasking and personality surveys. Table 2 shows a breakdown of the main behavioral measures analyzed (switch rate, return cost, interference cost, and *Popup$_{select}$*) here. Figure 3 shows correlational plots between the main behavioral measures analyzed and MMI score. Because we took a hierarchical regression modeling approach, we conducted three separate post hoc power analyses on the main analyses described using G*power (*Faul et al., 2009*), one for each separate model added.

*Switch rate.* The hierarchical model predicted switch rate (the percentage of trials in which a participant switched tasks across all trials in which a popup occurred) only in step 3

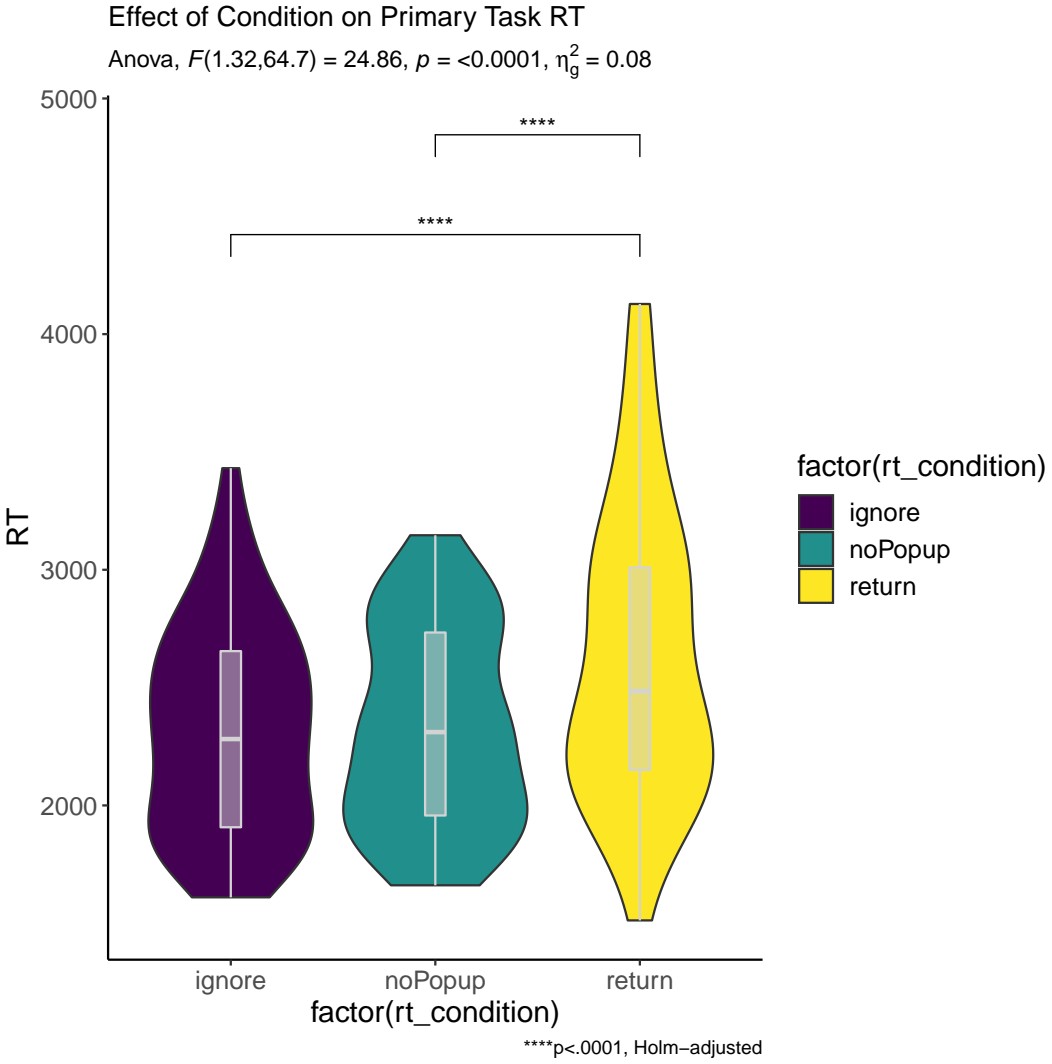

Figure 2 **Effect of condition on primary task RT.** Asterisks (****) denote a significance value of $p. < .0001$.

(Full model: $F(3, 86) = 5.6$, $p = .001$, Adjusted $R^2 = .13$). Only step 3 of the model achieved greater than 80% power according to a post-hoc power analysis. Individual predictors in the model were examined further, and only MPI score predicted switch rate ($B = 0.01$, $SE = 0.003$, $t = 3.8$, $p = <.001$). Given that the MPI is thought to reflect the tendency or preference to multitask, we expected MPI to relate to switch rate, which was supported by our results. Table S2 shows the hierarchical model at each step for this variable. To more directly relate MPI and switch rate, we correlated non-zero switch rate and MPI scores, and found a significant positive correlation ($r (65) = .33$, $p = .007$), suggesting that the tendency to multitask in day-to-day life, as indexed by the MPI, does indeed have at least a weak association with participants' choice to switch to the secondary task when given the opportunity.

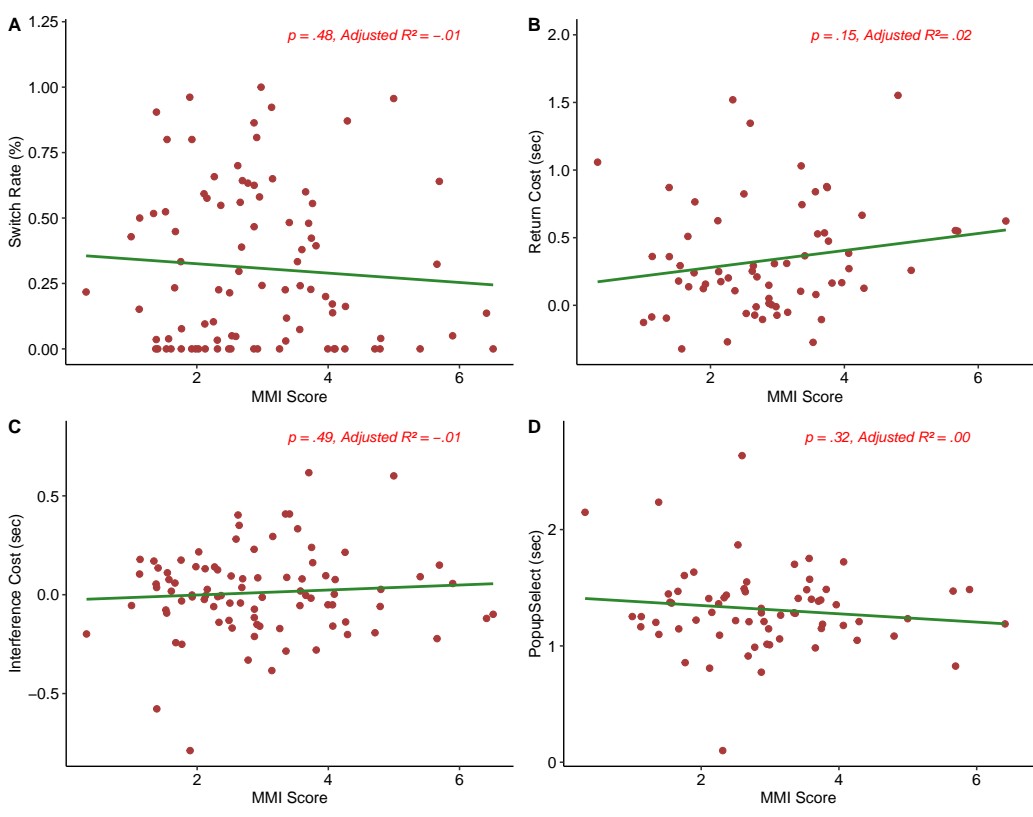

**Figure 3  MMI score *vs.* main behavioral measures (A–D).**

*Return cost.* None of the steps in the hierarchical regression model were significant for return cost, or the difference in average reaction time for primary tasks following a switch to the secondary task minus the average reaction time for all other primary task trials without a popup (Full model: $F(3,61) = 1.42$, $p = .25$, Adjusted $R^2 = .02$). A post hoc power analysis also suggested that we did not reach the sample size necessary to achieve above 80% power on any of the three steps of the model. We expected return cost to be related to MMI score, in line with previous work suggesting that media multitaskers show a decrease in task performance, but this was not the case. Table S3 shows the hierarchical model at each step for this variable.

*Interference cost.* The hierarchical model predicting interference cost, or the difference in average reaction time for primary task trials with a non-selected pop-up and average reaction time on primary task trials without a pop-up was not significant at any step (Full model: $F(3, 85) = 0.69$, $p = .56$, Adjusted $R^2 = -.01$). A post hoc power analysis also suggested that we did not reach the sample size necessary to achieve above 80% power on any of the three steps of the model. Table S4 shows the hierarchical model at each step for interference cost.

*Popup$_{select}$.* Table S5 shows the hierarchical model at each step for Popup$_{select}$. None of the three models predicting Popup$_{select}$, or the RT for participants to choose to switch after popup onset on relevant trials, were significant (Full model: $F(3, 63) = 0.93$, $p = .43$,

Adjusted $R^2 < .001$). A post hoc power analysis also suggested that we did not reach the sample size necessary to achieve above 80% power on any of the three steps of the model. Our pattern of results here suggests that there was no difference in the amount of time an individual took to elect to switch tasks in relevant trials in terms of degree of multitasking, impulsivity score, or preference for multitasking.

## Exploratory analyses

As this is a novel task with many components, we made several exploratory comparisons to examine the relationships between MMI and task performance. To this end we examined RT on trials in which the participant ignored the popup and completed the primary task (Popup$_{ignore}$), response time to elect to switch (*i.e.,* time to respond to the prompt, "A New Task is Available! Press 'Y' to switch tasks") Popup$_{select}$, response time on non-popup trials (Primary$_{nopopup}$), RT on primary trials following a switch, regardless of availability of a switch (Primary$_{return}$), RT on trials in which the individual repeated the primary task (Primary$_{repeat}$), and overall average RT on the primary and secondary tasks were also determined for each participant. As before, because some participants did not switch tasks at all, some of these measures could not be calculated for the entire sample. We again used a hierarchical multiple regression analysis to develop a model predicting each of these measures based on survey results. The first step of the model added MMI score to the model, while steps 2 and 3 added attentional impulsivity scores and MPI scores, respectively. Table S6 shows a breakdown of the exploratory analyses described here. Additionally, we compared individuals who did not switch at all during the task to those who did on each task and survey measure (where possible) using both parametric and non-parametric $t$-tests (where appropriate, as some task measures were non-normally distributed) to examine for any differences in task performance between both groups. Figure 4 shows correlational plots between the exploratory behavioral measures analyzed and MMI score. We again note that the current study did not achieve the sufficient statistical power needed to detect the weak effect of media multitasking on task performance, which may explain the pattern of effects found.

*Survey Results.* A Wilcoxon Signed-Ranks test suggested that there was no difference in MMI scores between those who did not switch at all throughout the task (median = 2.52) and those who did (median = 2.87) $W = 847$, $p = .76$. There was a significant effect for switch group, $t(59.9) = -3.29$, $p = .001$, indicating that those who did not switch at all had a lower MPI score($M = 33.4$, $SD = 8.17$) than those who did switch ($M = 40.5$, $SD = 11.27$). A Wilcoxon Signed-Ranks test suggested that there was an effect of switch group on the attentional impulsivity sub-scale of the BIS, $W = 562.5$, $p = .02$, between those who did not switch at all throughout the task (median = 32) and those who did (median = 38). These results suggest that both attentional impulsivity and preference for multitasking are positively related to the act of switching throughout the task in the current study.

*Primary$_{return}$.* Table S7 shows the hierarchical model at each step for this Primary$_{return}$. Step 1 of the model ($F(1, 63) = 4.1$, $p = .048$, Adjusted $R^2 = .05$) found that MMI score predicted Primary$_{return}$ ($B = 0.15$, $SE = 0.072$, $t = 2.02$, $p = .48$). Steps 2 and 3 were not significant (Full model: $F(3,61) = 1.47$, $p = .23$, Adjusted $R^2 = .02$). Our finding in step 1

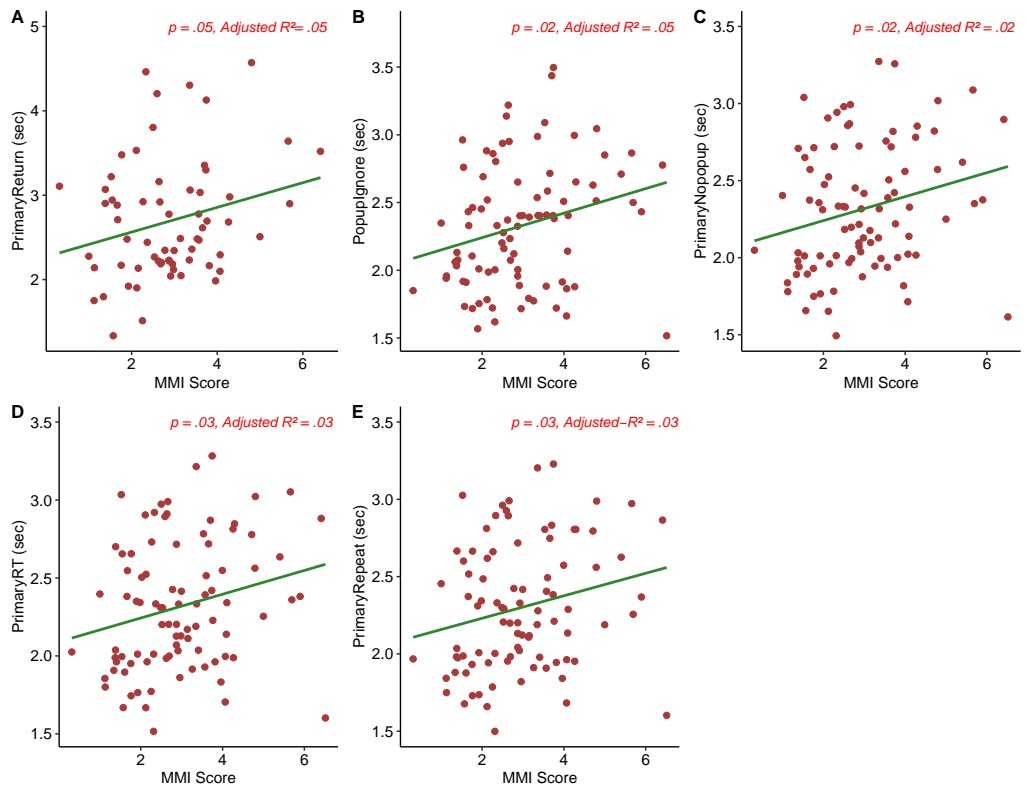

**Figure 4** **MMI score *vs.* exploratory behavioral measures (A–E).**

of the model suggests that those who media multitask more often show a decrease in their ability to return to an initial task following a switch in task set, such that they are slower to respond to the initial task regardless of the availability of a switch on that given trial. The lack of a relationship between MPI score and Primary$_{return}$ also suggests that those who are more likely to choose to multitask do not show an increase in performance when switching back and forth between task sets.

*Popup$_{ignore}$*. Table S8 shows the hierarchical model at each step for Popup$_{ignore}$ *i.e.*, the reaction time on trials in which a popup occurred but was not attended to. The initial step in our hierarchical model predicting Popup$_{ignore}$ from MMI score was significant $(F(1, 87) = 6.08, p = .02, R^2 = .05)$. MMI score positively predicted popup interference in the form of a longer RT, $(B = 0.09, SE = 0.037, t = 2.47, p = .02)$. Step 2 in the model was also significant $(F(2, 86) = 3.15, p = .048,$ Adjusted $R^2 = .05)$, but the change in $R^2$ was not. The step 3 model was also significant $(F(3, 85) = 4.28, p = .007,$ Adjusted $R^2 = .1)$, as was the change in $R^2$. A relationship between MMI score and Popup$_{ignore}$ RT suggests that the more an individual media multitasks, the slower they are on trials in which they decide to ignore popups, in line with the suggestion that heavy media multitaskers have difficulty filtering irrelevant information (*Cain & Mitroff, 2011*; *Ophir, Nass & Wagner, 2009*). The relationship between MPI score and Popup$_{ignore}$ suggests that although individuals may have a preference for multitasking, they may still be unable to

filter out irrelevant information during a task. There was no significant effect for switch group, $t(56.6) = -1.395$, $p = .018$, despite those who switched during the task ($M = 2.36$, $SD = 0.48$) having a longer RT than those who did not switch at all ($M = 2.23$, $SD = 0.37$) on trials in which a popup occurred but was ignored.

*Primary$_{nopopup}$*. Table S9 shows the hierarchical model at each step for this variable. The initial model predicting response time on tasks in which there was no popup prompt (Primary$_{nopopup}$) from MMI score was significant ($F(1, 88) = 5.32$, $p = .02$, Adjusted $R^2 = .05$). MMI score positively predicted RT on primary task trials with no popup prompt ($B = .08$, $SE = 0.034$, $t = 2.31$, $p = .02$). Step 2 of the hierarchical model was not significant, but step 3 was (Full model: $F(3, 86) = 3.34$, $p = .02$, Adjusted $R^2 = .07$). In this model, MPI score positively predicted RT on primary task trials with no popup prompt ($B = .008$, $SE = 0.004$, $t = 1.99$, $p = .05$), as did MMI score ($B = .08$, $SE = 0.03$, $t = 2.5$, $p = .01$). The relationship found between MMI score and Primary$_{nopopup}$ suggests that individuals who media multitask more often are generally slowed, while the relationship between MPI score and Primary$_{nopopup}$ suggests that preferential multitaskers show this same pattern of effects. A Wilcoxon Signed-Ranks test suggested that there was no difference in RT scores on trials in which no popup occurred between those who did not switch at all throughout the task (median $= 2.18$) and those who did (median $= 2.33$) $W = 660$, $p = .17$.

*Primary RT*. The initial step in the model predicting response time on all primary trials from MMI score was significant ($F(1, 88) = 5.16$, $p = .03$, Adjusted $R^2 = .04$). MMI score positively predicted Primary RT ($B = 0.08$, $SE = 0.034$, $t = 2.27$, $p = .08$). Step 2 was not significant, but step 3 was, ($F(3, 86) = 3.41$, $p = .02$, Adjusted $R^2 = .08$). The individual predictors in the model were examined further, and both MPI score ($B = 0.01$, $SE = 0.004$, $t = 2.08$, $p = .04$) and MMI score ($B = 0.08$, $SE = 0.03$, $t = 2.49$, $p = .01$) positively predicted Primary RT. Higher MMI and MPI was associated with slower RT on the primary task, suggesting an overall slowing for heavier media multitaskers, as well as those who prefer to multitask in general. Table S10 shows the hierarchical model at each step for this variable. A Wilcoxon Signed-Ranks test suggested that there was no difference in RT scores on trials in which the participant completed the primary task on the preceding trial between those who did not switch at all throughout the task (median $= 2.2$) and those who did (median $= 2.33$), $W = 664$, $p = .18$.

*Secondary RT*. All three steps of the models predicting RT on the secondary task were not significant (Full model: $F(3,60) = 0.26$, $p = .85$, Adjusted $R^2 = <.001$). Table S11 shows the hierarchical model at each step for *secondary RT*.

*Primary$_{repeat}$*. Table S12 shows the hierarchical model at each step for Primary$_{repeat}$. Finally, the first step in the model predicting RT on Primary$_{repeat}$ trials, or trials in which the participant completed the primary task on the preceding trial, from MMI score was significant ($F1, 88) = 4.69$, $p = .03$, Adjusted $R^2 = .04$). MMI score positively predicted ($B = 0.07$, $SE = 0.034$, $t = 2.17$, $p = .03$) reaction times on trials in which the primary task was also completed on the preceding trial. Step 2 was not significant, but the final step, which added MPI score to the hierarchical model, was (Full model: $F(3,86) = 3.1$, $p = .03$, Adjusted $R^2 = .07$). MPI score positively predicted ($B = 0.008$, $SE = 0.004$, $t = 2.04$, $p = .04$) reaction times on trials in which the primary task was also completed on the

preceding trial, as did MMI score ($B = 0.08$, $SE = 0.03$, $t = 2.39$, $p = .02$). This pattern of results suggests that those who media multitask as well as prefer to multitask more often are slower when attending to the same task for a prolonged period of time.

A Wilcoxon Signed-Ranks test suggested that there was no difference in RT scores on trials in which the participant completed the primary task on the preceding trial between those who did not switch at all throughout the task (median = 2.21) and those who did (median = 2.28) $W = 698$, $p = .30$.

## DISCUSSION

In this study we investigated the effects of self-reported media multitasking exposure on performance in a novel multitasking paradigm. This paradigm consisted of a primary and secondary task. Occasionally, during the primary task, a 'popup' prompt would appear, allowing the participant to switch tasks of their own volition. If they chose to engage with it, they would then complete a different secondary task before returning to the primary task. Participants were not pre-selected for extreme degrees of media multitasking as in many previous studies; we took an individual difference approach using naïve participants. We used hierarchical regression models to predict task performance based on self-reported media multitasking exposure and preferences and the Attentional impulsivity subscale of the BIS-11.

We hypothesized that media multitasking exposure (MMI score) and preference (MPI score) would predict both the frequency at which participants would elect to switch to the secondary task (*switch rate*), as well as the RT to choose to switch on relevant trials ($Popup_{select}$). We also expected a "*return cost*" and an interference cost that would be positively predicted by media multitasking scores. In addition to these initial constructs of interest, we also performed several exploratory analyses between the survey constructs and several other behavioral measures. These included the effect of each survey measure on the RT on primary task responses following a task switch ($Primary_{return}$), primary task RT on trials where a pop-up was presented, but the secondary task wasn't chosen ($Popup_{ignore}$), RT on trials in which no popup was present ($Primary_{nopopup}$), RT on primary task trials in which the participant completed the primary task on the preceding trial ($Primary_{repeat.}$), and overall RT on the primary and secondary tasks.

We found mixed results. In line with our primary hypotheses, we found that MPI score predicted switch rate. However, we found no significant predictors of return cost. Several exploratory analyses yields results supporting the hypothesis that media multitasking exposure relates to poorer executive function; we found that MMI score positively predicted $Primary_{return}$, $Popup_{ignore}$, $Primary_{nopopup}$, $Primary_{repeat}$, and primary RT. We also found that MPI score positively predicted $Popup_{ignore}$, primary RT, $Primary_{nopopup.}$, and $Primary_{repeat}$. Attentional impulsivity scores on the BIS-11 subscale were not substantial, with the exception of when comparing results across individuals who did not switch at all *versus* those who did. After comparing the survey scores of individuals who did not switch at all throughout the task *versus* those who did, we also found that those who switched tasks had higher MPI and attentional impulsivity scores than those who did not switch

tasks at all. Although we found several significant models and predictors, it is crucial to underline the fact that the effect sizes for all findings were small and as such are likely not indicative of any greater underlying trend. In fact, according to a post-hoc power analysis using G*power (*Faul et al., 2009*), the current study did not achieve the sufficient statistical power needed to detect the weak effect of media multitasking on task performance.

Supporting the idea that high media multitaskers show less efficient executive functioning (*Becker Alzahabi & Hopwood, 2013*; *Cain & Mitroff, 2011*; *Murphy & Creux, 2021*; *Ophir, Nass & Wagner, 2009*; *Sanbonmatsu et al., 2013*), we found that the greater an individual's MMI and MPI scores were, the greater their RT on primary trials following a switch was, regardless of a popup being present. This suggests that media multitasking reduces one's ability to re-engage with the primary task. This has been demonstrated in applied domains such as multitasking while driving (*Nijboer et al., 2016*; *Strayer, Watson & Drews, 2011*). However, our results regarding return cost, or the difference in RT on primary trials with no popup available following a task switch, may contest this interpretation. In regard to return cost, we found no effects within the three steps of our model. This may suggest that although heavier media multitaskers are less effective when switching back to a task from a previous task set on average, this difference is not detectable when only taking into account primary task RT on trials following a switch in which another switch is not possible. Additionally, both MMI and MPI score predicted overall RT on all primary task trials. Here, those who media multitask more often, as well as those who prefer to multitask, responded to the primary task more slowly in general. These results point toward a general decrease in task performance for individuals who engage in media multitasking more often.

Heavy media multitaskers have been found to have an inability to efficiently filter out distractors (*Lui & Wong, 2012*; *Murphy & Creux, 2021*; *Ophir, Nass & Wagner, 2009*). Our results suggest a similar relationship, with individual MMI score predicting reaction time during Popup$_{ignore}$ trials, or trials in which a popup occurred but the participant chose not to switch, such that responses to the primary task during these trials were slower for individuals who media multitask more often. In trials in which the participant chose not to switch, the popup can be seen as a distraction from completing the primary task. As such, a longer RT to complete the primary task here demonstrates an inability to effectively filter irrelevant stimuli to the task at hand. This pattern of effects is true regarding MPI score as well, suggesting that even preferential multitaskers may be distracted to a greater extent by a popup stimulus, even if they choose to ignore the option to switch tasks. However, our findings regarding interference cost, or the difference in RT on trials in which the participant ignored a popup and the RT on trials in which no popup occurred, may conflict with this interpretation. The lack of a relationship here may be attributed to the low amount of overall switches, which are further elaborated on below. The weak effect size associated with the former finding may also account for this discrepancy.

Because greater impulsivity and worse inhibitory control have been linked to MMI scores (*Gorman & Green, 2016*; *Sanbonmatsu et al., 2013*; *Murphy & Creux, 2021*; *Rogobete, Ionescu & Miclea, 2021*; *Shin, Webb & Kemps, 2019*), we expected a greater switch rate among more impulsive and less inhibited individuals. Despite this, we found no evidence that attentional impulsivity as indexed by the sub-scale of the BIS score predicts switch rate.

We did, however, find that individuals who did not switch tasks at all had lower attentional impulsivity scores than those who did, along with a lower preference for multitasking. We found similar results for the RT for individuals to choose to switch tasks (popup$_{select}$). We reasoned that more impulsive individuals would switch tasks more quickly and frequently, again because of the greater possibility of reward due to completing the secondary task more often.

Several factors may have contributed to the low switch rate observed in the current study ($\sim$31%). For example, the popup prompts may not have been salient enough to entice a switch. Increasing the points earned for completing the secondary task or making the popup more prominent on the screen by changing the text color or including sound may make the popups more salient. Because there was no monetary incentive for a higher score other than the motivation to "beat" a "high score", participants may have had no motivation to maximize points earned, leading to less task switches. This is a limitation of the task we must acknowledge, as we have no way to be certain that this did not affect our participants' motivation and thus, our results.

Relatedly, the greater penalties for an error in the secondary task may have also disincentivized task switches. Interestingly, higher MPI score was related to switch rate, but MMI score was not. Individuals who switched tasks had higher MPI scores when compared to those who did not switch at all as well. These effects point towards a greater propensity for preferential multitaskers to opt to switch to a different task set, but not for individuals who report engaging in media multitasking to a greater degree. Finally, there was a non-trivial difference in overall participant accuracy between the primary task (93%) and the secondary task (58%); one possibility is that participants found the secondary task too difficult and not worth the increased effort (*Inzlicht, Shenhav & Olivola, 2018*). This was not analyzed further due to the even greater potential for incomparable data due to the overall low switch rate observed.

The changes we made to the Media Use Questionnaire may have also contributed to some of the findings, both null and significant, in this experiment. Many of the changes made to the original 2009 questionnaire devised by Ophir and colleagues were done to reflect changes in the media consumption landscape we see today. Nevertheless, our average MMI score was relatively in line with other studies that have used the original *Ophir, Nass & Wagner (2009)* questionnaire. Despite this, we must still acknowledge that the changes made in the current study to the original questionnaire may limit the generalizability of our findings to other studies that used the original version. Since the original introduction of the media use questionnaire in 2009, there have been attempts to devise a more cohesive and brief version of the questionnaire, with differing patterns of effects (*Baumgartner et al., 2017*; *Pea et al., 2012*). This lack of uniformity in regard to screen time and media use measures in the overall literature points to a bigger problem recently emphasized by Kaye and colleagues (*2020*). They point out that the conceptualization of media use is far too broad and ambiguous in its current state in the literature, and vastly undermines the generalizability of the literature to a broader audience.

Despite the limitations discussed, the findings resulting from this novel multitasking paradigm are promising. Because the majority of current media multitasking literature has

used paradigms designed to evaluate other domains of executive function such as working memory and inhibitory control, the implementation of a paradigm specifically designed to be analogous to the environment in which individuals frequently multitask is needed. This initial study serves as a first step to fill this gap in the literature. Further implementations to this paradigm to develop a task more analogous to real world multitasking should include a sound clip in conjunction with the popup notification. This would be reminiscent of many of the notifications we receive on our phones and laptops, as they too may sometimes include sound. Many of the notifications we receive on these same devices can be ignored as they are "spam" or of little interest to us. As such, the inclusion of uninformative or "distractor" popups mixed in with informative popups may serve to further emulate a real-world multitasking environment. It may be beneficial to include trials in the paradigm where a task switch is required to allow for a clear differentiation between an interference cost and trials in which a participant actively chooses to switch costs, as in our current design, this is not possible. Finally, a larger sample size is needed to provide for enough statistical power to detect effects our paradigm may uncover.

## CONCLUSIONS

Using a novel, more ecologically valid paradigm, we expected to find a negative effect of media multitasking, multitasking preference, and attentional impulsivity on task performance. We found a number of significant, effects of media multitasking on task performance, including a general slowing effect on the primary task. We also found that self-reported multitasking preference related to how often participants chose to engage in the secondary task. These findings contribute to the now growing media multitasking literature showing some of the negative effects of frequent media multitasking. However, it is crucial to recognize that many of the effects we found were weak, and because of our a smaller than ideal sample size, may not persist given further testing. Further, the adjustments made to the Media Use Questionnaire may limit the generalizability of our findings to previous work using the original questionnaire. Future directions of this line of research include a modification to the paradigm to make the popup prompts more "enticing" to participants to more closely mirror a real-world multitasking environment. We also plan to collect EEG data to examine the event related potentials occurring as participants complete the tasks.

### Funding

Funding for this research was provided for by the Texas A&M Department of Psychological and Brain Sciences. The Texas A&M Office of Graduate and Professional Studies also funded Jesus J. Lopez through the Diversity Fellowship. The funders had no role in study design, data collection and analysis, decision to publish, or preparation of the manuscript.

### Grant Disclosures

The following grant information was disclosed by the authors:
The Texas A&M Department of Psychological and Brain Sciences.
The Texas A&M Office of Graduate and Professional Studies.

### Competing Interests

The authors declare there are no competing interests.

### Author Contributions

- Jesus J. Lopez conceived and designed the experiments, performed the experiments, analyzed the data, prepared figures and/or tables, authored or reviewed drafts of the paper, and approved the final draft.
- Joseph M. Orr conceived and designed the experiments, prepared figures and/or tables, authored or reviewed drafts of the paper, and approved the final draft.

### Human Ethics

The following information was supplied relating to ethical approvals (i.e., approving body and any reference numbers):

The Texas A&M University Institutional Review Board granted ethical approval to carry out the study with its facilities (IRB2018-1456M).

### Data Availability

The raw data files, as well as the R and JASP files/scripts, and codebooks are available in the Supplemental Files and at OSF:

Lopez, Jesus, and Joseph M Orr. 2021. "Effects of Media Multitasking Frequency on a Novel Volitional Multitasking Paradigm." OSF. August 13. doi: 10.17605/OSF.IO/NJU8A.

### Supplemental Information

Supplemental information for this article can be found online at http://dx.doi.org/10.7717/peerj.12603#supplemental-information.

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
