# Peer review of "Effects of media multitasking frequency on a novel volitional multitasking paradigm"

_PeerJ, doi:10.7717/peerj.12603_

## Round 0.1 · original submission · Major Revisions

I have received thoughtful reviews from two experts in the field. I thank them for their time and effort. While both are enthusiastic about the topic of your study, both also have series reservations about your reporting of the experiment that will necessitate major revision of the manuscript before it meets threshold for publication at PeerJ. Their critiques appear below, but I would like to highlight a few of the most critical observations here.

As a researcher whose work attempts to increase the ecological validity of our laboratory techniques, I applaud your motivations in the current work. However, I am also mindful of the fact that sensitivity usually decreases as we move away from highly-controlled (and oft-validated) laboratory techniques toward those that are more representative of life beyond the lab. Reviewer 2 notes that your framing of the experiment, including your choice of title, is likely to confuse the reader and potentially misrepresent your goals with the piece. If we want to explore variability in executive function across a continuum of self-professed preferences for multitasking, then a classic measure of executive function (one that has been specifically designed and refined to detect variability in executive function) is a useful tool and will likely be far more sensitive to variability than a task with more moving parts that better emulates the messiness of lived experience. If there is reason to believe that the “standard” measure is not capturing meaningful variability, then that argument should be explicitly developed, but I do not think that was what you were intending. Further, if you believe that your new technique is more sensitive to variability in executive function or simply more valid, then that argument also needs to be better developed.

Reviewer 2 also notes that your title does not adequately represent the experiment. I concur with this observation. Your new task is meant to be a “real-world” test of multitasking or task switching, yet in the title the task is framed as artificial (i.e., “lab-based”). Meanwhile, differences in general media multitasking are being assessed in the standard way, via self-report measures, not via some kind of in-vivo observation of tendencies toward multitasking, as the title implies. Please revise your title to better represent what you are studying, and in a way that does not imply that a causal claim can be made from the data.

While the literature review is generally good, it is lacking acknowledgment that claims about the relationship between screen time and cognitive function tend to be exaggerated and based on vanishingly small effects. Reviewer 2 points you to Orben and Przybylski (2019). In recent years, Amy Orben has produced scads of work to support this talking point. I would also point you to Kaye et al. (2020), on which Orben was an author.

Throughout the manuscript, both reviewers (and myself) noted a need for enhanced clarity. Many methodological details are missing or are handled only tersely. For example, as noted by Reviewer 1, more detail is needed on the scoring techniques for the BIS, and additional justification is needed for deviations from the original protocols for the MMI, since your changes complicate comparison with the prior literature. Furthermore, you need to justify your sample size. How was a sample size of 90 established?

Your reporting of Results needs a major overhaul. As Reviewer 2 notes, it would be more helpful to the reader to include your tables among supplementary materials and visualize your meaningful outcomes in one or two figures. It is laudable that you share all of your data, but a third party would not be able to make sense of what you have shared. Please revise these materials to be more clearly annotated. An additional codebook document would be especially helpful. One small point that neither reviewer seemed to note: Your body text says that “Table 1 shows a breakdown of survey scores by gender.” It does not (and should not) depict the gender breakdown.

Generally, I request that you add a statement to the paper confirming whether you have reported all measures, conditions, data exclusions, and how you determined your sample size. You should, of course, add any additional text to ensure the statement is accurate. This is the standard reviewer disclosure request endorsed by the Center for Open Science [see http://osf.io/project/hadz3]. I include it in every review.

The remainder of the critiques below are clear. I believe they can be addressed in a revision, but likely necessitate a major reworking of the manuscript. I look forward to reading a revision on this work.

References:

K. Kaye, L., Orben, A., A. Ellis, D., C. Hunter, S., & Houghton, S. (2020). The conceptual and methodological mayhem of “screen time.” International Journal of Environmental Research and Public Health, 17(10), 3661. http://dx.doi.org/10.3390/ijerph17103661

Reviewer 1 ·

Basic reporting

The manuscript examines the application of the human behaviour of media multitasking to a real-world task that employs media multitasking skills such as cognitive flexibility/task switching. The authors have provided a clearly written overview of the study in much of the manuscript. There are a few areas where the clarity of the information presented could be improved.

The authors have provided a suitable overview of the relevant literature relating to media multitasking and cognitive functions. However, several new papers have been published within this area since 2017 including a 2021 review and it would be relevant for the authors to include some of these newer pieces of evidence within the introduction and discussion sections of the manuscript.

The structure of the article is clear and follows the required journal article conventions. Raw data has been provided in accordance with journal requirements.
It is recommended that Figures 2 and 3 be removed from the manuscript as they do not provide significant value add to the information already presented within the manuscript. Figure 1 is generally clear in providing an overview of the sequence within a trial of the experiment. However, a key is needed to ensure clarity around which responses the key presses represent. For example, is it C = Correct and I = Incorrect? This needs to be clarified in Figure 1.

The authors have provided a comprehensive account of this experiment within the manuscript. The proposed hypotheses stated within the introduction and detailed within the results section are suitable and do follow from the literature review provided. However, there is some further clarification required within the study predictions. It would seem that hypotheses 2 and 3 are stating the same thing? This may not be the case, but rewording would ensure greater clarity. Specifically, this comment applies to this section of the manuscript on lines 154 to 160. “We also expected that participants would show a “return cost”, i.e., respond slower to return to the primary task following a switch to the secondary task that would be negatively predicted by media multitasking. Additionally, we predicted that individuals who media multitask more often would choose to switch to the secondary task more quickly and would have more trouble returning to the primary task after a switch, in line with the suggestion that media multitasking frequency is associated with decreased executive function (Baumgartner, Weeda, van der Heijden, & Huizinga, 2014; Cain et 161 al., 2016).”
It is also unclear exactly how this prediction relates to the concept of impulsivity. This needs to be unpacked to ensure clarity. Lines 161 to 164. “In line with the suggestion that frequent multitaskers show difficulties with filtering or impulsivity, we predicted that MMI score would also show a positive relationship with the amount of interference exhibited on trials where a pop-up was presented, but the secondary task wasn’t chosen.”

One issue that does require modification in relation to hypotheses is the exploratory analyses. While I agree that these are suitable analyses to conduct within the discussion section the outcomes of the exploratory analyses are frequently referred to in relation to predictions or hypotheses being supported or the results of this analysis being consistent with the prediction. Given that this is an exploratory additional analysis, and it does not appear to have specific predictions as would be the case the discussion section needs to be reworded when referring to the outcomes of the exploratory analysis. This revision is required for all instances of the discussion section where this reference is made. (Lines 422 to 605).

Experimental design

The authors have clearly identified the relevant gap within the empirical work on media multitasking research. That is, the area is heavily reliant on standard measures of cognition especially executive functions and has not really examined the link between media multitasking behaviour and performance on a more real world like task that examines cognitive flexibility/task switching.
Statements throughout the manuscript to illustrate good adherence to required ethical clearance standards for the protocol.
The experiment has been well designed and the measures outlined within the methods section of the manuscript. There are a few areas of the method section that could include additional details to ensure full replication of the experiment was possible or to provide greater clarity regarding decisions made about the included measures.
1) Was the sample size of 90 participants of sufficient power to detect the often-small effects evident in media-multitasking studies? Some clarification about power is required within the manuscript.
2) All the forms of media included within the MMI need to be stated within the method section detailing this measure (lines 177 to 205).
3) Within the information presented in lines 177 to 193 a more detailed explanation is required as to why the scale used for participants’ responses differed from the standard MMI measures (line 184 to 205). The current statement to get a more precise measure does not seem to be a sufficient statement to justify this decision. For example, how and why would this 5-point scale provide more precision and also allow cross study comparison compared to the standard 4-point scale that is used within this measure?
4) Sample items are required for the Multitasking Preference Inventory as well as a clear statement of the scale terms used for the Likert scale.
5) How was the BIS scored and which scales or sub-scales were used in this study and why? What were the anchors used within the Likert scale for the BIS? Key details are missing for this measure.
6) It is unclear why participants would have wanted to switch to the secondary task (lines 223 to 235). Since there was no reward for doing this other than points and it does not seem the points were of use for any reason then why would participants undertake the secondary task at all. Explain how this design decision would not have adversely affected the results of the study in any way.

Validity of the findings

All participant data has been provided with the manuscript submission, thank you.
Clarification is needed to justify the use of the total BIS score within the analysis instead of sub=scale scores.
Lines 291 to 293 note that the MMI mean score is in line with that obtained from a prior study. However, the mean for that study was considerably lower than has been reported in most other studies. Therefore, there needs to be justification as to why this difference in MMI mean scores is suitable and this should be supported with more than one other study used as an example. What method was used to compute the MMI values in this study and is it the same or different to that used in most other studies within this area?
Table 1 requires a note below that has a key to indicate what each abbreviated measure refers to exactly.
Table 2 requires more detail about what the exact measures were that were used on the analysis. The current title is not informative in relation to what factors were included. This same comment applies to Table 6 as well.
Check the z value reported on line 311 as this does not seem to be correct.
The DVs (units of analysis) for the main analysis for the Switch Rate, Return Cost and Interference Cost need to be clearly stated within the text. It is not clear how these variables were measured (e.g., percent correct, percent errors, response time etc.).
In lines 476 to 478 a comment is made the repeating the same task and the link with MMT is explained by attention rather than any other cognitive process such as working memory, inhibitory control (staying on task rather than allocating attention to extraneous thoughts). This needs further clarification to fully justify this point.
Line 492 to 494 suggest that the results could be due to weak effects sizes. There needs to be clear indication that this cannot be attributed to the sample size for this study not being sufficient to examine the possible weak effects that are evident in media multitasking studies examining a link with cognitive function.

Additional comments

Lines 461 to 463. Check expression for sentence starting with “At the same time…”
Lines 559 to 562, check expression in sentence starting with “It is also important to note…”
Was there any analysis conducted between those who undertook both tasks compared to those participants who did not multitask at all for the BIS, MMI or MIP? This analysis could be very informative regarding differences or lack there of between these two groups of participants. This analysis would further add to the overall strength of the study and could be included within the exploratory analysis section of the manuscript.
Check results section as some minor typos (e.g., brackets missing to end brackets) are evident.
Overall this is a very interesting study that informs the literature regarding the link between self-reported media multitasking and cognitive performance.

Reviewer 2 ·

Basic reporting

The idea to assess the relationship between media multitasking habits and voluntary task switching in a setting that somewhat mimics a real-world scenario is of great interest. However, the way the authors frame and introduce their study is rather confusing.

For example, the abstract states that media multitasking has been linked to decreased executive functioning followed by a critique that this finding has been established using “standard” tasks rather than a real-world volitional multitasking environment; which justifies the current study. I must admit that I don’t understand this line of reasoning. First, to use standard executive functions tasks to evaluate the relationship between media multitasking habits and executive functions is a valid strategy in my opinion. Second, if the point is to argue that their multitasking paradigm is a more valid measure of executive functions, this point should be substantiated in the text.


The paper is overall hard to read and lacks precision. For example, the title “real-world media multitasking shows few effects on lab-based volitional multitasking performance” seems to indicate that the authors assessed “real-world” media multitasking (as opposed to the “classic” way) when in fact media multitasking was assessed using the typical questionnaire and the “real-world” aspect refers, in the main text, to their new multitasking task paradigm rather than the media multitasking. Furthermore, the title seems to imply a causal relationship, which is not substantiated by the data.

The multitasking test developed by the authors is in my opinion a misnomer since participants do not perform both tasks at the same time but are required to switch between tasks. I understand the authors may have a different view on these terms (multitasking == task switching) but if so, they should define them.

The coverage of the literature could be improved to be both more to the point and substantive. For example, I don’t see the value of describing “supertaskers” in the context of this study (line 104) and I missed information that there is evidence showing that the link between digital media consumption and various outcomes is very small (e.g., Orben & Przybylski, 2019). There have also been studies investigating the relationship between media multitasking and multitask performance (e.g., using the operation span task, Sanbonmatsu et al 2013; cited in the paper line 711). It would be useful to explain in greater detail how that task differs from the one presented here and discuss the results accordingly.

The reporting of the results is also confusing and lacks focus. Instead of displaying 13 tables, the authors might display 1-2 figures that emphasise the specific points they want to make; those tables would be more adequately placed in the supplementary material, in my opinion.

Finally, although the data has been shared, it is hard to make sense of that data. No codebook or explanation is provided and it seems that the data is somewhat inconsistent. For example, within file 063process.xls, the “participant” variable takes on values “switchrate”, “totalswiches” but also “63”. More work needs to be done for this dataset to be usable by other researchers. The osf directory also contains a bit of code; but it’s unclear how it relates to the results reported in the paper (it’s not even clear which data is being used for the analyses reported in the paper); providing the code that was used for the analysis in this paper would also help understand what exact analyses were conducted.

Experimental design

The paper seems to be within the scope of the journal (in particular as there have been other papers in cognitive psychology published in PeerJ).

As described in the previous section, the research question has not been presented in a clear and compelling way. I believe the overall design to be interesting but currently poorly justified.

I like the idea of their volitional task-switching design but I’m also a bit worried about two aspects. a) it’s unclear to me why some participants decided not to switch at all to the secondary task since this task provided more points. This behavior may perhaps indicate that some people did not engage with the task, did not understand how it worked or simply did not pay attention to the notifications—in which cases the interpretation of the results would be very different. Perhaps a better way to engage participants would have been to offer them the possibility to quit the experiment as soon as they reach a set level of points (in which case abstract points, which some may not value, would translate to time, which most people value).
b) it seems to me that it would have been necessary to also include trials where task switching was required; this would allow for instance to separate “interference costs” due to the processing of the “pop up” (i.e., detecting information on the screen) versus that related to deliberating whether or not to switch tasks.

Validity of the findings

As stated earlier, the data is provided in a way that is not directly usable and the code used to run the analyses reported in the paper has not been shared, as far as I can tell. Hence, it is not possible for me to evaluate the robustness and soundness of the reported analyses.

The validity of the findings is also unclear for reasons already stated in the previous sections.

---

## Round 0.2 · Minor Revisions

Thanks for your thoughtful response to the reviews. I believe the manuscript has improved substantially, but there are some lingering issue that need to be addressed before the manuscript meets threshold for publication. Reviewer 1 from the first round of review generously agreed to provide feedback on your revision. Their feedback below is quite clear and should be easily addressed.

I concur with the reviewer that the manuscript is still rather verbose and occasionally hard to follow. The Results section is dense. This cannot be remedied, but your Discussion and Conclusion sections can and should be tightened up. The reviewer questions your use of the term "two-sided" as it relates to a correlation analysis. Certainly "two-tailed" is a more common description, but I have seen both terms used. I will leave it up to you which terminology fits most naturally with your voice.

I also ask that you work through the manuscript with a fine-toothed comb to resolve lingering typos and clarity issues. I noted the following:
- line 86: The sentence starting "As a result..." is unclear. It currently reads as if heavy media multitaskers are filtering out...their lighter media multitasking counterparts.
- Line 168: use parentheses around the citation rather than brackets
- Line 234: "istening" should be "'listening"
- Line 322: remove first instance of "trials"
- Line 325: Should you be referring to MEAN reaction times here?
- Line 376: It might be good to replace "condition" with "factor"
- Line 381: Scientific notation for small p-values gives a false sense of precision. Please replace with "p < .001".
- Line 398: Capitalize "Individual"
- Line 410: Again, I presume you should be invoking MEAN reaction time.
- Line 417: Again, I think this is about a difference score on mean reaction times.
- Line 475: Remove first instance of "show"
- Line 610: "serious" should be "series"
- Line 690: Delete first instance of "related to"

Reviewer 1 ·

Basic reporting

The structure of the article is clear and follows the required journal article conventions. Raw data has been provided in accordance with journal requirements.
The authors have provided a comprehensive account of this experiment within the manuscript.
The overall logic of the study and situating the study within the literature and relevant context has been improved from the previous version of this manuscript.
The clarity of the written document is suitable.

Experimental design

The additional method information provided within the revised manuscript has addressed my comments from the prior version of the manuscript.

The MMI formula alone in Sup Figure 1 is not informative as to how the calculation is actually done this needs to be revised
The BIS measure cannot be included within the supplemental figures as this is a published scale and this would be a copyright issue if it is included without permission. Further it is not needed for this article. This same comment applies to the MPI in Sup Figure 3
Supplementary Figure 2 which is a copy of the MMI is not needed the method information in the revised manuscript now clarifies things around that measure

Validity of the findings

Please check the wording of a two-sided Pearson correlation it is not clear what this actually means. Does it mean two-tailed or something else.
Why were non-parametric tests used for the multitasking performance analysis? This has less power than a parametric test and justification is needed for use of the test.
Figure 5 the correlation should be presented as a Table as that is easier to follow than the Figure currently presented.
Comment on line 578-79 that effects may no longer exist with a larger sample size does not make sense please clarify what is meant by this.
In general I think the length of the discussion section could be reduced. Many of the points made while important could be made more directly and more concisely and still convey the same message for the reader.
This same comment of reducing the word length applies to the conclusion section. State the main aim and then cover the key significant and non-significant findings. Then focus on the key take home message from the article. This will help improve readability of the last sections of the manuscript.

Additional comments

line 610 serious should be series

---

## Round 0.3 · accepted · Accept

I did not send the manuscript out for further review. I have read it and your response to previous reviewers. I am satisfied that the reviewers' critiques have been addressed, and I am happy to accept this manuscript for publication in PeerJ!